# Compartment specific responses to contractility in the small intestinal epithelium

Taylor Hinnant [1,2⦿], Wenxiu Ning [1,3⦿]*, Terry Lechler [1,2]*

**1** Department of Dermatology, Duke University Medical Center, Durham, North Carolina United States of America, **2** Department of Cell Biology, Duke University Medical Center, Durham, North Carolina United States of America, **3** Center for Life Sciences, School of Life Sciences, Yunnan Key Laboratory of Cell Metabolism and Diseases. Yunnan University, Kunming, China

⦿ These authors contributed equally to this work.
* wenxiu_ning@ynu.edu.cn (WN); terry.lechler@duke.edu (TL)

**Data Availability Statement:** All relevant data are within the manuscript and its Supporting Information files.

**Funding:** This work was funded by a grant from the National Natural Science Foundation of China to W.

## Abstract

Tissues are subject to multiple mechanical inputs at the cellular level that influence their overall shape and function. In the small intestine, actomyosin contractility can be induced by many physiological and pathological inputs. However, we have little understanding of how contractility impacts the intestinal epithelium on a cellular and tissue level. In this study, we probed the cell and tissue-level effects of contractility by using mouse models to genetically increase the level of myosin activity in the two distinct morphologic compartments of the intestinal epithelium, the crypts and villi. We found that increased contractility in the villar compartment caused shape changes in the cells that expressed the transgene and their immediate neighbors. While there were no discernable effects on villar architecture or cell polarity, even low levels of transgene induction in the villi caused non-cell autonomous hyperproliferation of the transit amplifying cells in the crypt, driving increased cell flux through the crypt-villar axis. In contrast, induction of increased contractility in the proliferating cells of the crypts resulted in nuclear deformations, DNA damage, and apoptosis. This study reveals the complex and diverse responses of different intestinal epithelial cells to contractility and provides important insight into mechanical regulation of intestinal physiology.

## Author summary

The small intestine epithelium is comprised of two main compartments: the villi which contain differentiated cells that function in nutrient absorption, and the crypts which are made up of undifferentiated cells which serve to replenish cells of the villi. Because of their physical location within the tissue, villi and crypts are subjected to different types of insults and mechanical forces. We sought to directly test how villi and crypts respond to mechanical changes in the epithelia by genetically inducing actomyosin contraction. Increasing contractility in villar cells resulted in cell shape changes without affecting their overall polarity or organization. However, it led to a non-autonomous increase in proliferation of the undifferentiated cells of the intestine. In contrast, increased contractility in

N. (No. 32270846), and from the National Institutes of Health to T.L. (R01-AR081081 and R01-DK117981). The funders had no role in study design, data collection and analysis, decision to publish, or preparation of the manuscript.

**Competing interests:** The authors have declared that no competing interests exist.

the proliferative cells of the crypt resulted in nuclear shape changes, DNA damage and ultimately rapid cell death. Thus, our work demonstrates that the crypt and villi epithelia respond differently to mechanical changes and highlights long-range regulation between villi and crypt compartments.

## Introduction

Cells sense and respond to multiple mechanical inputs to shape tissue, regulate cell fate, and maintain homeostasis [1–4]. These forces can be intrinsic, coming from the cells' own cytoskeleton, or they can be extrinsic, coming from external sources including extracellular matrix, neighbor cells, fluid shear forces, and intra-tissue tension. Intrinsic forces are largely derived from contractility of the actomyosin cytoskeleton, which, in epithelial tissues lie downstream of many signaling pathways and is under tight control of cell-cell or cell-matrix attachments [5,6].

The small intestine's lining is a simple epithelial sheet that is folded into distinct structures, the villi, and crypts. The villi project into the lumen of the gut and are populated with differentiated cells, largely enterocytes, that play important roles in digestion and barrier formation. Conversely, the crypts descend into the stromal layer of the small intestine and contain proliferating stem and transit amplifying cells. These cells fuel the rapid turnover of the intestinal epithelium which is replaced every 3–5 days [7–9]. Coordination between differentiated cells and stem cells within the intestinal epithelium is required for gastrointestinal homeostasis and regeneration [10,11]. While pathogenic infections and other types of physiologic insults leading to villi barrier disruption cause progenitor responses, the underlying pathways mediating crosstalk between these cell compartments are not fully understood.

The intestine is exposed to many physical inputs from digested food, extracellular matrix/basement membrane, smooth muscle contractions driving peristalsis, and deformations of the bowel wall [12]. Prior work indicated that epithelial cells respond to these mechanical inputs to regulate their proliferation, differentiation and motility [13,14]. For example, distention of an isolated small bowel segment using lengthening devices increased both crypt depth and villus height [15,16]. Moreover, intrinsic forces generated from cytoskeletal remodeling are required for both the morphogenesis of intestinal compartments as well as the maintenance of this folded structure [17,18]. Regulated F-actin dynamics are required for active villar cell migration, driving the homeostatic flux of the tissue [19]. Additionally, actomyosin is required for both cell extrusion, which is important for villar cell turnover, and cell contraction in response to infection [20,21]. Despite the importance of actomyosin-mediated contractility in intestinal development and homeostasis, and its modulation during pathologic conditions, the effects that contractility has on cell and tissue function remain unclear. This is largely due to the fact that very few tools exist to specifically tune a cell's contractile state *in vivo*, in order to study its cell and tissue level effects. By manipulating contractility *in vivo*, in the absence of additional perturbations, we hope to define the cell and tissue-specific responses that are induced by increased acto-myosin activity.

In this study we genetically induced increased actomyosin activity to probe the role of cell contractility in the two epithelial compartments of the small intestine. Strikingly, we find very different responses to local contractility. Increasing contractility in differentiated villar epithelia cells locally changes cell shape and drives non-cell autonomous hyperproliferation of the transit amplifying cells of the crypt. This hyperproliferation response leads to increased cell flux through the crypt/villar axis. In contrast, crypt cells are highly sensitive to autonomous increases in contractility, leading to nuclear deformation, DNA damage, and apoptosis. This

study highlights distinct, compartment specific responses to cell contractility and, for the first time, the extreme sensitivity of progenitor cells in the small intestinal crypt to intracellular contractile forces.

## Results

### Generation of a mouse model to increase contractility in the villar epithelium

In order to control contractility in the intestinal epithelium, we have taken advantage of a construct, Arhgef11$^{CA}$, that acts as a constitutively active version of a guanine-nucleotide exchange factor (GEF) for RhoA, thus promoting contractility [22]. This construct contains only the GEF region of Arhgef11 (DH/PH domains), which are fused to a membrane targeting sequence. Prior work in cultured cell lines demonstrated that optogenetic activation of this construct resulted in increased contractility in these cells [23]. We previously generated a mouse line, TRE-Arhgef11$^{CA}$, which allows inducible expression of this transgene [24]. This transgene caused visible cell contraction when expressed in epithelia monolayer culture. It additionally lead to whole-tissue contraction and an increase in junctional cytoskeletal proteins when expressed in mouse epidermis [24].

To induce contractility in the intestine, we mated TRE-Arhgef11$^{CA}$ to Villin-rtTA mice, allowing for doxycycline-inducible expression in the intestine epithelium (Fig 1A). As expected, administering doxycycline to *Villin-rtTA;TRE-Arhgef11$^{CA}$* mice induced Arhgef11$^{CA}$ expression in the small intestine, and the resulting HA-tagged protein localized to the cell membrane of differentiated villar cells (16.9% of villar cells). However, it was only rarely expressed within the crypt region and then only at the very top of these structures (Fig 1B and 1C) (only 0.5% of total crypt cells were HA positive, with less than 7% of crypts containing any HA positive cells; 147 crypts counted from 4 mice). Thus, we refer to this model as Arhgef11-$^{Villi}$. Analysis of tissue sections and whole mount staining of intestines two days after doxycycline induction revealed mosaic expression of Arhgef11$^{CA}$, with a higher density of HA+ cells near the base of villi compared to more sparse expression near the tops of villi (Fig 1B).

Co-staining of HA along with phalloidin to visualize F-actin revealed that there was no change in apical F-actin in Arhgef11$^{CA}$ -expressing cells after two days of expression (Fig 1D–1G). This is likely due to the initial high levels of F-actin that exist within the brush border. However, there was a significant increase in the intensity of lateral F-actin staining in Arhgef11$^{CA}$ expressing cells, compared to the low intensity of lateral F-actin in the control (Fig 1G). As myosin II binds F-actin and drives contractility, we next examined the localization of myosin IIC. Consistent with previous findings, myosin IIC was strongly localized to the apical surface in the intestinal epithelium (Fig 1H). However, lateral myosin IIC increased significantly in Arhgef11$^{CA}$-expressing villar cells compared with controls (Fig 1I and 1J), which was consistent with the increased lateral F-actin. There was also an increase in the junctional levels of myosin IIA (S1H Fig). We next assessed whether the increase in cortical actomyosin resulted in increased tension on adherens junctions, mechano-sensitive cell-cell adhesions that coordinate the underlying F-actin cytoskeleton. Tension induces conformational changes in the adherens junction component α-catenin, exposing an epitope that can by recognized by the α-18 antibody [25]. We noted a cell-autonomous increase in α-18 staining intensity in cells expressing Arhgef11$^{CA}$. Cells that were one cell distance away from the Arhgef11$^{CA}$-expressing cells had levels of α-18 similar to control junctions, suggesting that the contractility is not transduced across the epithelium, but has only local effects (S1A–S1F Fig). These data suggest that increased contractility results in local effects that are not transmitted more than one cell away from the contraction. Therefore, the Arhgef11$^{Villi}$ mouse line serves as an inducible genetic model of increased cellular tension specifically within intestinal villar cells.

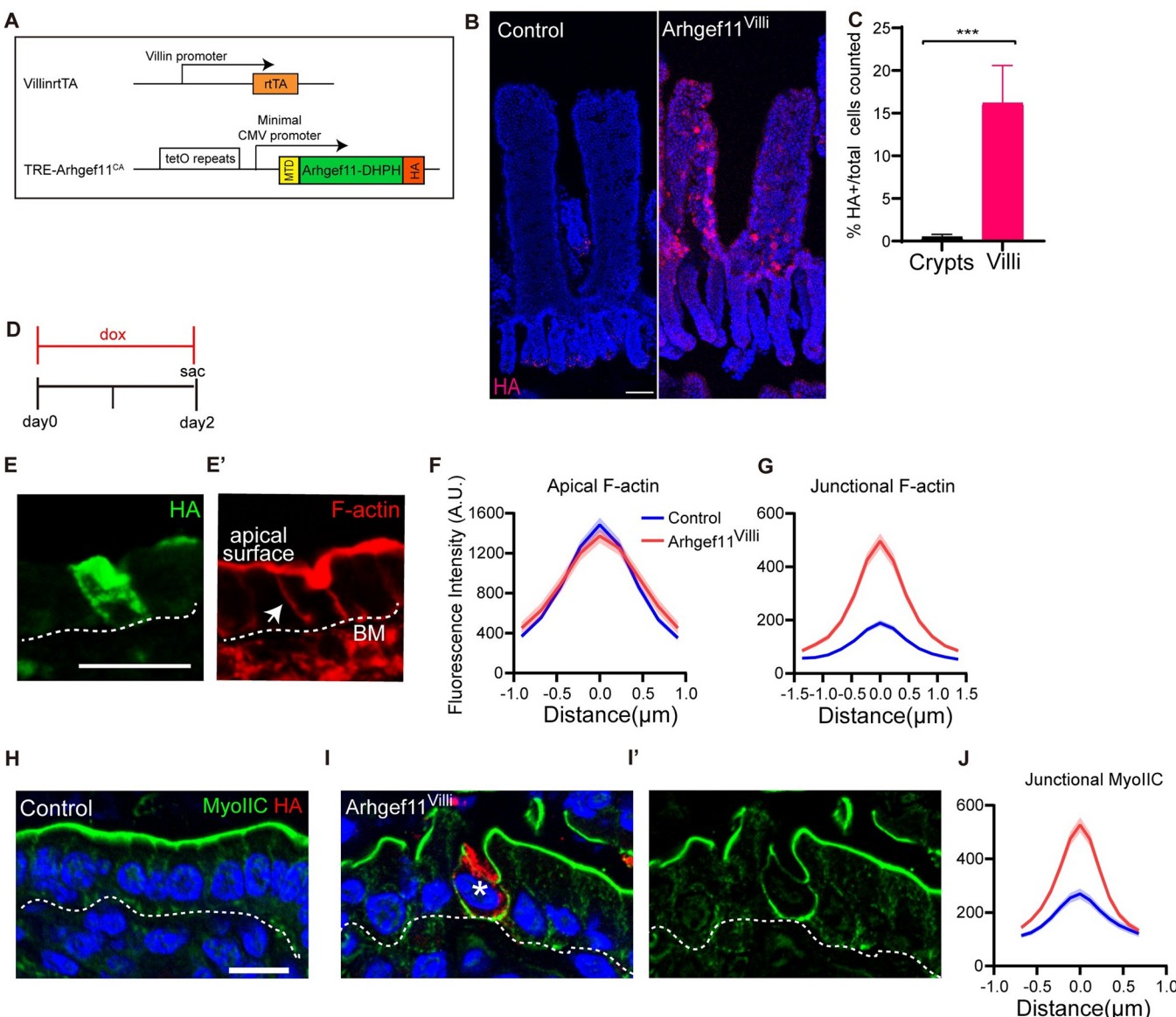

**Fig 1. Genetic induction of actomyosin contractility in the intestinal villi.** (A) Diagram of genetic scheme for expressing Tre-Arhgef11[CA] in the small intestine villar epithelia. (B) Whole mount immunofluorescent images of control and Arhgef11[Villi] intestinal epithelia stained with HA (red). Scale bar 50µm. (C) Quantification of the mean percentage of HA+ cells in crypts and villi. n = 3 mice (3 crypts or villi per animal) p = 0.0027, unpaired t-test. (D) Timeline of experimental procedure. (E-E') Immunofluorescent images of Arhgef11[Villi] stained with (E) HA (green) and (E') F-actin (red). Basement membrane marked with dotted line. Arrow marks lateral junction. Scale bar 20µm. (F-G) Quantification of (F) apical F-actin and (G) junctional F-actin intensity. Data are mean ±SEM. (F) n = 40 cells for control and n = 29 HA+ cells for Arhgef11[Villi]. p = 0.85, unpaired t-test. (G) n = 36 cells for control and n = 40 HA+ cells for Arhgef11[Villi]. p = 0.0058, unpaired t-test. (H-I) Immunofluorescence staining of myosin IIC (green) and HA (red). Basement membrane marked with dotted line. Asterisk marks HA+ cell. Scale bar 10µm. (J) Junctional myosin IIC fluorescence intensity of control or HA+ epithelial cells. n = 33 cells for control and n = 51 HA+ cells for Arhgef11[Villi]. p = 0.0153, unpaired t-test.

## Arhgef11[villi] induces cell autonomous effects on cell shape and non-autonomous effects on tissue architecture

When we stained for myosin IIC and F-actin, we observed that inducing expression of Arhgef11[CA] not only increased lateral F-actin, but also induced changes in cell shapes (Fig 1I). In control intestinal villi, the cells were uniformly columnar. However, Arhgef11[CA] expressing

cells were irregular in shape. This also resulted in cell shape deformations in immediately adjacent cells that were not expressing Arhgef11CA (S2A–S2D Fig). To determine whether these changes reflected alterations in the polarity of these cells, we stained for a panel of proteins with well-established polarity in intestinal epithelial cells. As already mentioned, the apical brush border is still strongly marked by F-actin staining (Fig 1E'). ZO-1, a tight junction marker, remained localized to the apical junctional region (S2G Fig). E-cadherin, a component of adherens junctions and a lateral marker, was also found along the lateral membrane and excluded from the apical region in cells expressing Arhgef11CA (S2H Fig). We further examined intracellular polarity and found that GRASP65, a Golgi marker, and pericentrin, a centrosome marker, were localized apical to the nucleus, as in control cells (S2E, S2F and S2I Fig). These data demonstrate that increasing contractility in villar epithelial cells affects cell shape but does not perturb organelle organization and cell polarity. That being said, we did note very rare cases where contractile cells had either internalized into the epithelium and thus lost their apical domain, or where they appear to have been extruded from the epithelium and lacked visible contact with the basement membrane.

Mice tolerated the expression of Arhgef11CA in the villi well, with no apparent gross defects or change in weight over the short term (S3A Fig). Consistent with this, we did not observe a significant difference in the length of the whole intestine (S3B Fig). To test if increasing actomyosin contractility in Arhgef11Villi affects epithelial morphology, we performed whole mount staining of the intestinal epithelial sheets. There was no difference in the length of villi in Arhgef11Villi (508.9±10.0 μm) compared to the control (508.8±14.4 μm) (Fig 2A–2C). However, the length of crypts was significantly increased in Arhgef11Villi (150.9±3.2 μm) versus control (101.9±2.1 μm) (Fig 2D–2F). This crypt elongation phenotype extended to the colon, where Arhgef11HA was expressed at the tops of crypts, causing a moderate increase in the length of colon crypts (S4A–S4C Fig). These data reveal an unexpected non-cell autonomous effect of contractility in differentiated cells of the villi on the proliferative crypts. This phenotype was fully reversible, as when mice were examined five days after removal of doxycycline from their diet, crypt length had returned to normal levels (S5 Fig).

## Increased contractility in villi promotes proliferation of progenitor cells in the crypt

To determine the cause of increased crypt length in the Arhgef11villi mice, we began by examining the proliferative status of the mutant tissue. Staining for a marker of actively cycling cells (Ki67) revealed a substantial increase in both the relative proportion and the absolute number of Ki67+ cells per crypt in the Arhgef11Villi mice (Fig 2G–2I). As Ki67 labels all proliferative cells, we next wanted to determine whether this reflected an increased pool of stem cells. We found that the number of stem cells per crypt, marked by Olfm4 expression, is similar in control and mutant small intestine (S6A–S6C Fig) [26]. There was also no change in the domain of Olfm4 expression measured from the base of the crypt (S6D Fig). In agreement with this data, there was no change in the stem cell pool marked by high Sox9 expression (S6E Fig) [27,28]. This suggests that stem cell numbers have not increased, or that a fraction have lost expression of stem cell markers. Regardless, these data indicate that the crypt hyperplasia, in response to increased actomyosin contractility in villar cells, is likely driven by transit amplifying cell hyperproliferation and not an expansion of stem cells. To determine whether the increased proliferation altered the cell fates of progeny, we stained for Mucin2, a marker of goblet cells. We found similar percentages of goblet cells in the villi of control and mutant mice, suggesting that while proliferation is increased, proper cell fate determination occurs (S7 Fig).

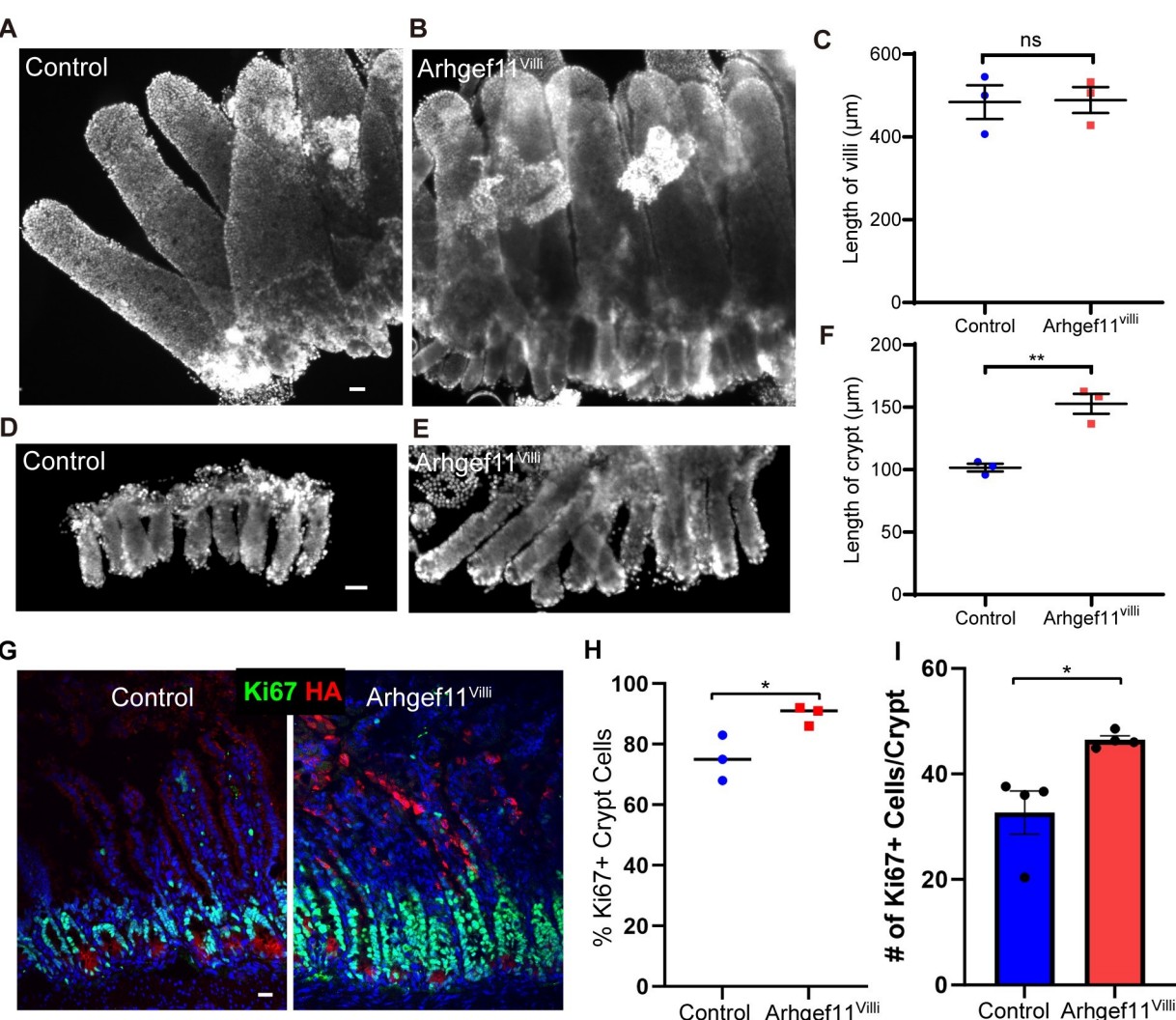

**Fig 2. Increasing contractility causes non-cell autonomous crypt hyperproliferation.** (A-B) Whole mount immunofluorescent images of control (A) and Arhgef11$^{Villi}$ (B) intestinal epithelia (villi and crypts) stained with Hoechst. Scale bar 40μm. (C) Quantification of villi length in microns. For control n = 3 animals (20 villi). For Arhgef11$^{Villi}$, n = 3 animals (41 villi). p = 0.9292, paired t-test. (D-E) Whole mount immunofluorescent images of control (D) and Arhgef11$^{Villi}$ (E) crypts stained with Hoechst. Scale bar 40μm. (F) Quantification of crypt length in microns. For control n = 3 animals (51 crypts). For Arhgef11$^{Villi}$, n = 3 animals (64 crypts). P = 0.0040, paired t-test. (G-I) Immunofluorescent staining of Ki67 (green) and HA (red) in control and Arhgef11$^{Villi}$ intestinal sections. Scale bar 20μm. (H) percentage of Ki67+ cells in control and Arhgef11$^{Villi}$ crypt cells, n = at least 200 cells per mouse from 3 animals for control and Arhgef11$^{Villi}$ p = 0.0384, unpaired t-test, and (I) average number of cells per crypt in control and Arhgef11$^{Villi}$, n = 4 animals for control (33 crypts) and Arhgef11$^{Villi}$ (32 crypts). p = 0.0333, paired t-test. Mice were exposed to doxycycline for 2 days prior to each analysis.

Previous lineage-tracing experiments demonstrated that crypt cells proliferate, differentiate, and move upward along the villi until their progeny reach the villi tips and are lost[7,29]. This whole process happens within 3–5 days [7,9]. To test if cellular flux was affected in the small intestine of Arhgef11$^{Villi}$ mice, we performed EdU pulse-chase experiments where proliferative cells were labeled with a single injection of EdU one day after transgene induction (Fig 3B and 3D). After either a one- or two-day chase, we noted that the extent of migration up the villi axis was greater in the Arhgef11$^{villi}$ mice than in control (Fig 3C). At two days, the EdU-progeny extended to the tops of villi in the Arhgef11$^{Villi}$ mice, but only midway in the controls (Fig 3E and 3F). Thus, the increased proliferation in the crypt drives both crypt elongation as well as increased flux through (though not increased length of) the villi.

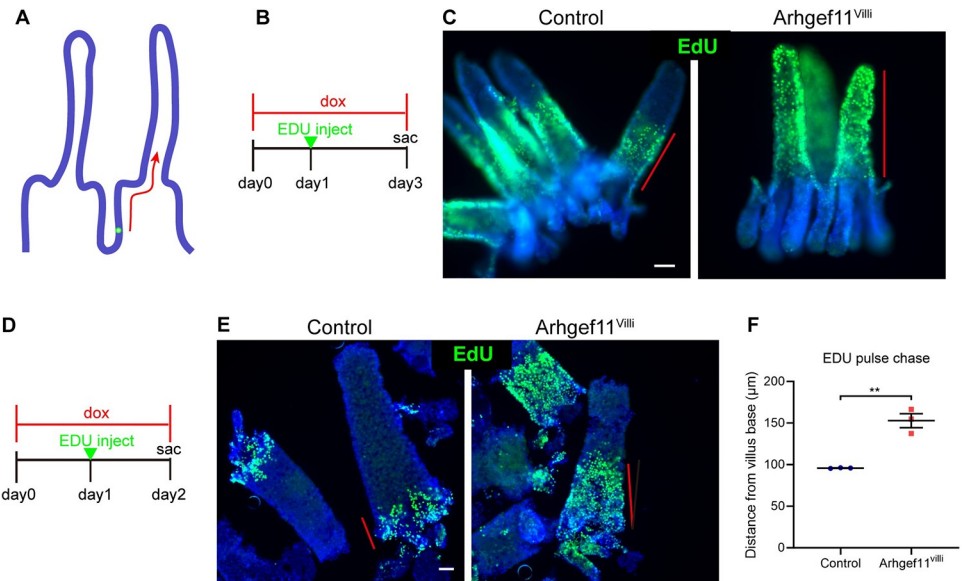

**Fig 3. Increased villar cell contractility speeds up intestinal cell transit time.** (A) Schematic of labeling method to measure cell transit rate. (B) Timeline of two-day EdU chase experiment. (C) Whole mount immunofluorescent images of control and Arhgef11$^{Villi}$ intestinal epithelia stained with EdU (green) two days after induction of label. Red lines depict the domain of EdU labeled cells. Scale bar 50μm. (D) Timeline of one day EdU chase experiment. (E) Whole mount immunofluorescent images of control and Arhgef11$^{Villi}$ intestinal epithelia stained with EdU (green) one day after induction of label. Red lines depict the domain of EdU labeled cells. Scale bar 40μm. (F) Quantification of the distance of the final EdU labeled cell from the base of villi in microns. For control n = 3 animals (27 villi). For Arhgef11$^{Villi}$, n = 3 animals (47 villi). P = 0.0025, paired t-test.

## Increasing contractility in crypts results in DNA damage and apoptosis

Given the non-autonomous effects of contractility on crypt proliferation, we next addressed whether there were direct effects of contractility on these cells. To test this, we generated Villin-Cre;Rosa-rtTA;TRE-Arhgef11$^{CA}$, with the expectation that this would drive expression throughout the intestinal epithelium. Surprisingly, however, we found that in this background, Arhgef11$^{CA}$ was specifically expressed in the intestinal crypts with no expression detected in the villi (84.4% of crypt cells, 0% of villar cells) (Fig 4A–4C). This allowed us to directly test the function of contractility without the complicating consideration of non-cell autonomous effects from villar contractility. We refer to these mice as Arhgef11$^{Crypt}$.

Expression of Arhgef11$^{CA}$ in the small intestinal crypts led to an increase of junctional F-actin (Fig 4G–4I), similar to our observations in villi (Fig 1E–1G). The increase in junctional F-actin was accompanied by cell shape changes within the crypt, as well as gross changes in crypt morphology (Figs 4G–4H, S8A, and S8B). The normally smooth and circular apical surfaces of the mid-crypt region were misshapen and jagged in the mutant, also leading to increased crypt lumen size (Figs 4H–4J, S8A and S8B). These data demonstrate that increased contractility in crypt cells leads to changes in both cell shape and crypt architecture.

On an organismal level, increased contractility had dramatic consequences on gross animal well-being. The Arhgef11$^{Crypt}$ mice showed a significant weight loss, and a decrease in the total length of the small intestine after just one day on doxycycline chow (Fig 4D–4F). This rapid decline in animal health and deterioration of intestinal structure precluded us from extending analysis past one day of Arhgef11$^{CA}$ induction in the crypt.

To begin to understand the underlying causes of these contractility-induced phenotypes, we first examined proliferation. Unlike Arhgef11$^{Villi}$ mice, Arhgef11$^{Crypt}$ mice had a decrease

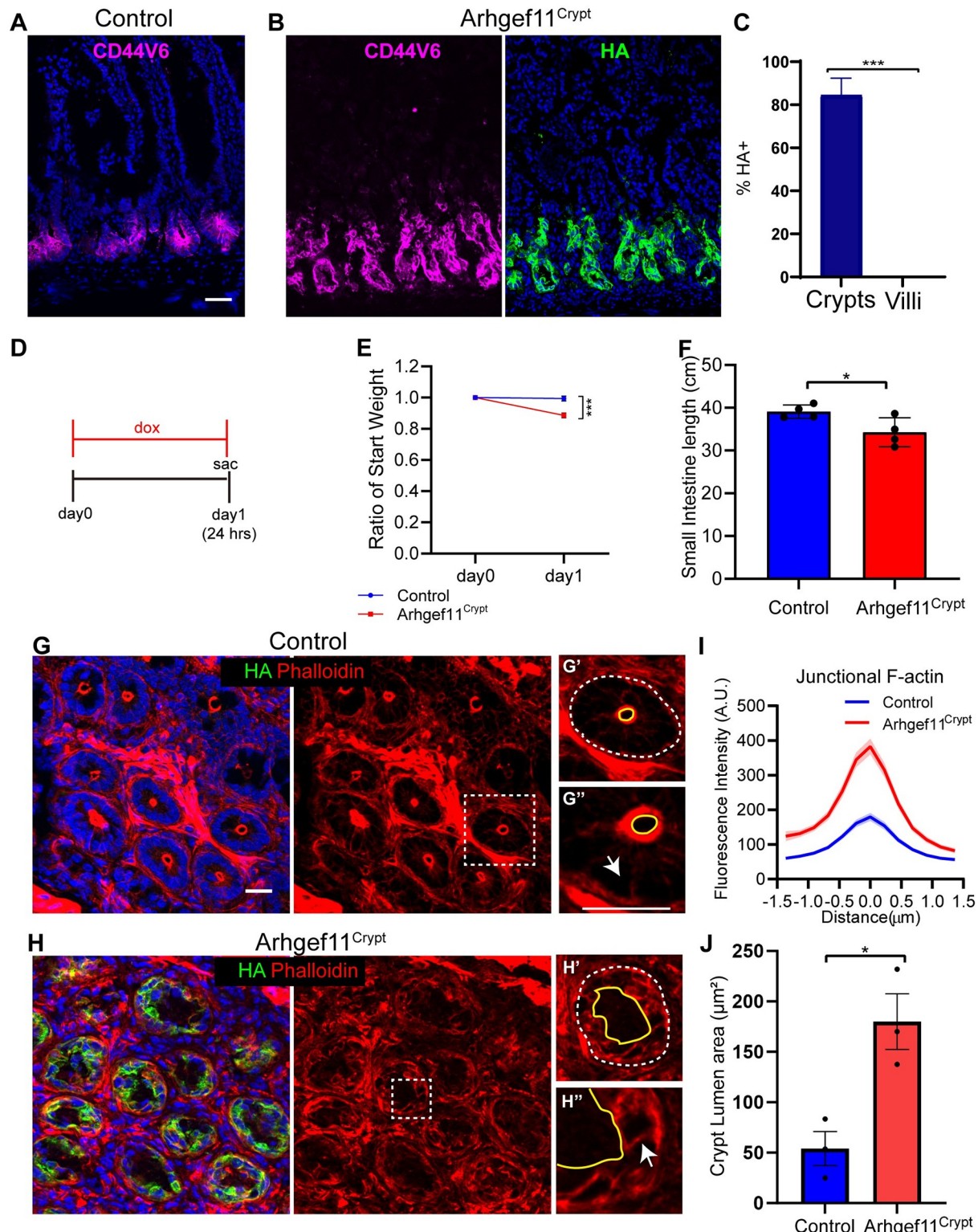

**Fig 4. Increased contractility in the crypt compartment leads to a rapid decline in organismal health.** (A-B) Immunofluorescence images of (A) control and (B) Arhgef11<sup>Crypt</sup> intestine sections stained with Cd44v6 (magenta) and HA (green). Scale bar 40μm. (C) Quantification of the mean percentage of HA+ cells in crypts or villi. n = 3 mice (3 crypts or villi per animal) p<0.0001, unpaired t-test. (D) Timeline of experimental procedure. (E) Average ratio of weight change after one day of doxycycline induction for control n = 3 animals and Arhgef11<sup>Crypt</sup> n = 3 animals. p = 0.0009, unpaired t-test. (F) Quantification of small intestine length of control and Arhgef11<sup>Crypt</sup> animals in

cm. (G-H) Immunofluorescence images of (G) control and (H) Arhgef11Crypt crypt cross sections stained with phalloidin (red) and HA (green). Scale bar 20μm. Squares denote insets for single crypt cross sections. (G' and H') Crypt lumen marked with yellow line and denotes apical side of crypt cells. The dashed white line marks basal side of crypt cells. (G" and H") Arrows mark lateral junctions. Scale bar 20μm. (I) Quantification of junctional F-actin fluorescence intensity of control or HA+ epithelial cells. Data are mean ±SEM. n = 27 cells for control and n = 27 HA+ cells for Arhgef11Crypt animals. p = 0.004, unpaired t-test. (J) Average crypt lumen area, n = 30 crypts from 3 animals for each genotype. p = 0.0267, paired t-test.

in the percentage of Ki67 positive cells in the crypts of the small intestine (Fig 5A–5C). Further, there was a decrease in the total cell number in each crypt (Fig 5D). Additional analysis of crypt morphology revealed that they were longer and thinner, demonstrated by an increase in crypt aspect ratio (Fig 5E). We observed that there was abundant apoptosis within the crypts of the mutant mice, as indicated by active caspase-3 staining (Fig 6A and 6B). This high rate of cell death and loss of crypt regeneration likely explains the defects in crypt architecture and decline in intestinal function.

We also noted that nuclei from the crypts of Arhgef11Crypt mice were misshapen (S8C and S8D Fig). Notably, nuclear deformations can promote DNA damage in proliferating cells [30]. This observation led us to examine whether these cells show evidence of DNA damage. Indeed, we found a massive increase in staining for the double stranded DNA break marker γ-H2AX in Arhgef11CA-expressing crypts, which was present in both Olfm4+ stem cells and transit amplifying populations (Fig 6C–6E). Quantitation revealed higher levels and earlier induction of DNA breaks as compared to apoptosis. The DNA damage phenotype could be partially rescued through inhibition of Rho kinase (Rock) signaling. Arhgef11Crypt mutant mice injected with the Rock1/2 inhibitor Y-27632 [31] 8 hours after initial transgene induction and assessed after 24 hours later showed decreased levels of γ-H2AX expression compared to untreated Arhgef11Crypt mice (Fig 6F–6G).

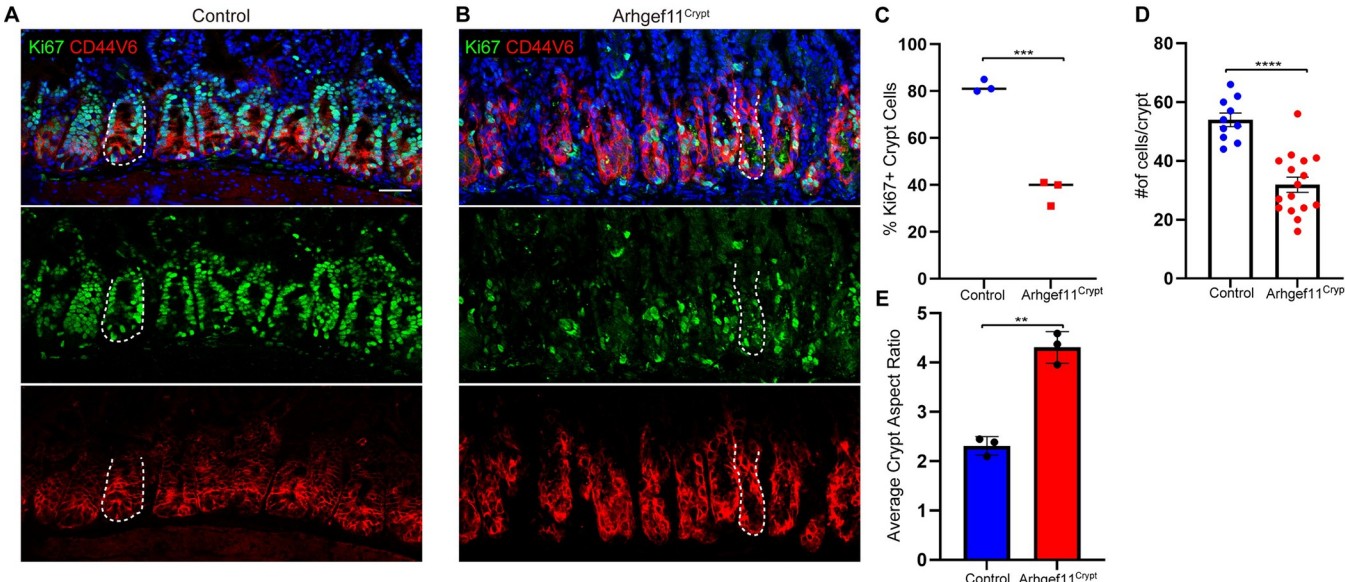

**Fig 5. Increased crypt cell contractility leads to changes in crypt architecture and a decline in proliferation.** (A-B) Immunofluorescence images of (A) control and (B) Arhgef11Crypt intestine sections stained with Cd44v6 marking crypt cells (red) and Ki67 (green). Dotted line marks a single crypt. Scale bar 40 μm. (C) Average percent of Ki67+ cells for control and Arhgef11Crypt n = at least 200 cells per mouse from 3 animals. p = 0.0002, unpaired t-test. (D) Average number of cells per crypt in cross section for control n = 10 crypts from 3 animals Arhgef11Crypt, n = 16 crypts from 3 animals. p<0.0001, unpaired t-test. (E) Average crypt aspect ratio for control n = 33 crypts from 3 animals Arhgef11Crypt, n = 37 crypts from 3 animals. p = 0.0017, paired t-test.

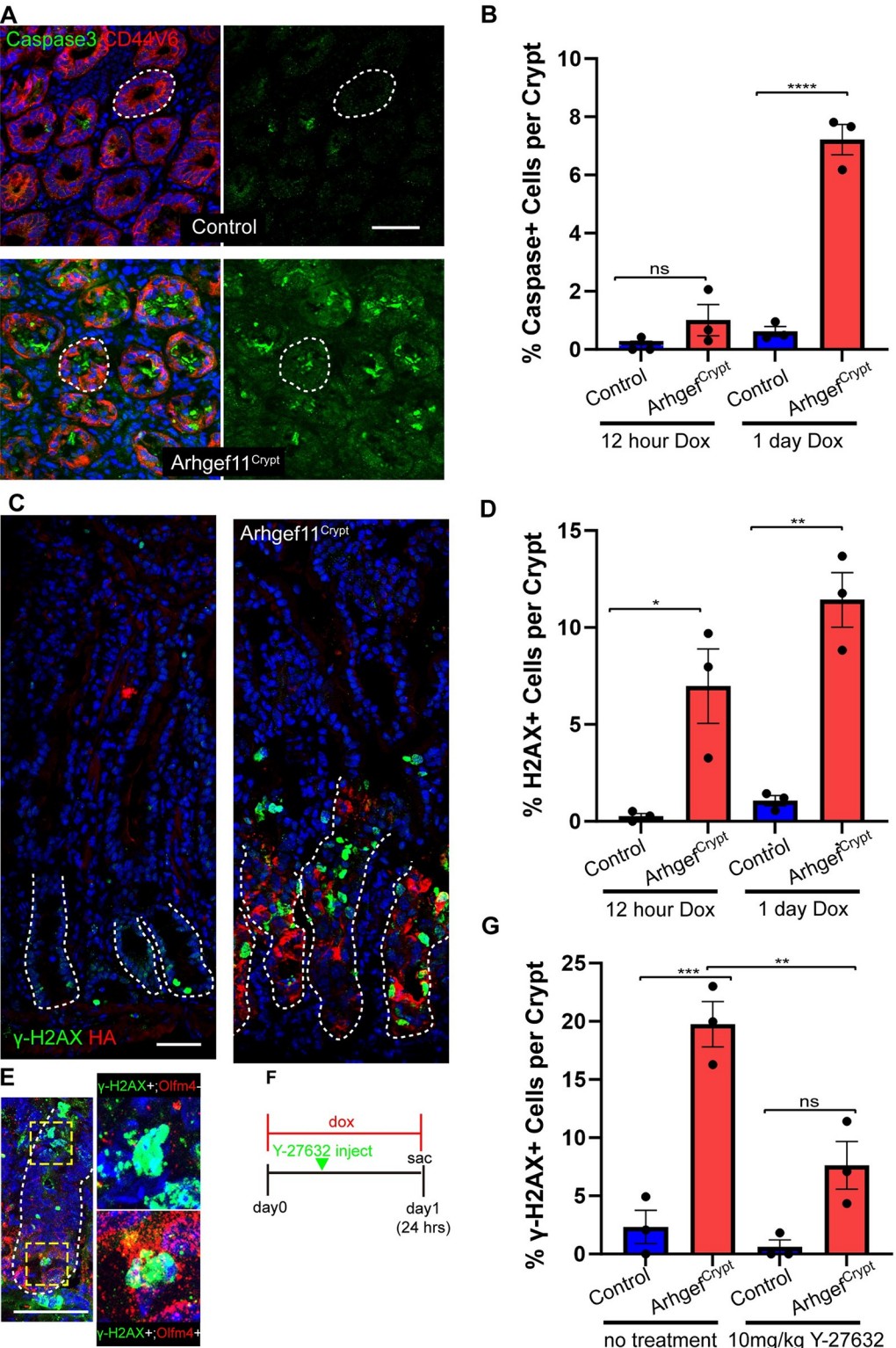

**Fig 6. Increased crypt cell contractility induces DNA damage.** (A) Staining of active-Caspase3 (green) and Cd44v6 (red) in control and Arhgef11<sup>Crypt</sup> crypt cross sections. Scale bar 40μm. (B) Quantification of percentages of caspase positive crypt cells after 12 hours or one day of dox induction. n = at least 7 crypts from 3 animals each for control and Arhgef11<sup>Crypt</sup>. p = 0.443 for 12 hour and p<0.0001 for 1 day. Tukey's multiple comparisons test. (C) Staining of γ-H2aX (green) and HA (red) in control and Arhgef11<sup>Crypt</sup> sections. Scale bar 40μm. (D)Quantification of percentages of γ-H2aX

positive crypt cells after 12 hours or one day of dox induction n = at least 7 crypts from 3 animals each for control and Arhgef11$^{Crypt}$. p = 0.0176 for 12 hour and p = 0.0013 for 1 day. Tukey's multiple comparisons test. (E) Representative crypt stained for Olfm4 (red) and γ-H2aX (green) in an Arhgef11$^{Crypt}$ sample. Insets depict γ-H2aX+ cells in the upper part of the crypt (top) and at the crypt base (bottom). Scale bar 50μm. (F) Timeline of Rock inhibitor rescue experiment. (G) Quantification of percentages of γ-H2aX positive crypt cells after one day of dox induction in Y-27632 treated and untreated mice. n = 3 animals (34 crypts) for control untreated, 3 animals (44 crypts) for Arhgef11$^{Crypt}$ untreated, 3 animals (35 crypts) for control Y-27632 treated, 3 animals (33 crypts) for Arhgef11$^{Crypt}$ Y-27632 treated. p = 0.0003 for no treatment and p = 0.0601 for Y-27232 treated. p = 0.0032 for Arhgef11$^{Crypt}$ untreated versus Y-27632 treated. Tukey's multiple comparisons test.

In addition, we found that DNA damage was positive in cells that were not positive for active caspase-3. These data suggest that the DNA damage is not a consequence of apoptosis. We did not observe γ-H2AX positive cells in the Arhgef11$^{Villi}$ intestines (no double positive HA+, γ-H2AX+ cells were observed, n = 98 cells from 3 mice), demonstrating that the crypt cells are more sensitive to increased actomyosin contractility. This phenotype extended to progenitor cells of the colon as well. Analysis of the colon revealed an upward trend in the level of DNA damage, albeit at variable levels, upon induction of cell contractility (S4D–S4F Fig). Together these data demonstrate that mechanical deformation induces DNA damage and apoptosis of progenitor cells, which leads to rapid organismal decline.

## Discussion

This study uncovered distinct effects of increased contractility in different intestinal epithelial cell types. We found that activated contractility in differentiated cells of intestinal villi led to local cell shape changes without effecting overall cell polarity. It also caused non-autonomous changes to the stem cell compartment through an increase in crypt depth and cell number. In contrast, proliferating intestinal cells within the crypt were highly sensitive to increased contractility, which led to nuclear deformations, DNA damage and cell death. Grossly, this resulted in rapid weight loss of the animals.

While cell autonomous cell shape changes were an expected effect of increasing contractility in villar cells, the resulting changes in tissue architecture, especially increased crypt length, were surprising. This is superficially similar to what is observed in the epidermis–contraction of differentiated suprabasal cells resulted in increased proliferation of basal progenitor cells [24]. The underlying mechanism of these non-cell-autonomous effects are currently unknown in the intestine. The changes in contractility appear to be quite local and we have no evidence that they are transmitted through multiple cells. Further, the response is rapid, within 18 hours of doxycycline induction of contractility, with the protein expression first observed at about 6 hours. We did not see changes in tight junction protein localization or increased cell death via cleaved caspase-3 staining, suggesting that contractile cells do not disrupt the epithelial barrier via cell death or loss of adhesions.

We are currently unable to rule out non-autonomous effects of villar cell contractility, as we have found that the compartmentalized expression of Arhgef11$^{CA}$ is lost when intestinal organoids are cultured from Arhgef11$^{Villi}$ animals. This leads to rapid organoid death, precluding us from analyzing epithelial only effects. Additionally, while immune infiltration often leads to a hyperproliferative response [32–35], we did not see any changes in immune cell content (macrophages and T-cells) in the intestinal epithelium. However, we cannot rule out immune signaling of resident populations. We often observed that the transgene was expressed only at the base of growing villi (Fig 1B), and therefore cannot rule out that contractile cells are being lost due to live-cell extrusion, though we only infrequently observed cells that appeared to be extruding or extruded. This raises the possibilities that contractility-dependent secretion of

paracrine factors from villar cells or the loss of villar cells via extrusion promotes proliferation of progenitors.

In contrast to the non-cell autonomous effects of inducing contractility in differentiated populations of the small intestine, induced contractility of stem and TA populations caused drastic local effects on cell function and resulted in rapid weight loss. Increased contractility in these populations induced DNA damage in the crypt compartment, likely causing the cessation of cell division and subsequent cell death. This leads to fewer cells per crypt and changes in crypt architecture. Previous studies have shown that restricting and deforming the nucleus of dividing cells causes DNA damage during replication [30]. As we observe deformed and elongated nuclei upon the induction of cell contraction in the crypt, it is possible that these forces are being transduced to the nucleus and causing these mitotically active cells to undergo replicative DNA damage. Notably, we do not observe nuclei deformities or DNA damage when inducing contractility in the postmitotic cells of the villi, suggesting that nuclear/cytoskeletal connections may be distinct in these two compartments.

The tissue and organismal level effects we observe upon increased contractility bring up many questions regarding why villar and crypt epithelial cells would need to respond differently to changes in cell contractility. For example, pathogen infection prompts both bacterial and epithelial-cell mediated myosin activation. It has been well documented that pathogens appropriate epithelial cell Rho-GTPase pathways upon their infection to aid cytoskeletal mediated invasion into the infected epithelial cells [36,37]. In addition to this, a study in murine intestinal epithelial cells showed that cells elicit prompt contractions in response to *Salmonella* infection that is independent of the bacteria-mediated effectors of actomyosin activation [20]. In each of these cases, the small intestine must mount a response to protect against the threat of pathogenic infection, balancing the expulsion of infected cells with the replacement of new cells. Therefore, it is tempting to speculate that in addition to responses mounted by the immune system to facilitate pathogen infection [35,38], that the small intestinal epithelial tissue itself responds to changes in tissue/cell contraction by increasing crypt cell proliferation. Through these mechanisms, the tissue could more quickly slough off infected cells and replace them. However, it is unclear what long-range signals are triggered by villar cells upon cytoskeletal activation to elicit this response. Interestingly, our data point to a rapid response to stem and TA cell activation, as the crypt depth increases in as short as 18 hours after contractility is induced, and quickly reaches a steady state that does not further extend even upon long term villar cell contraction. It is also noteworthy that crypt hyperproliferation does not always result in increased crypt depth [39], raising interesting questions about how crypt and villar length are set.

Finally, it is interesting to compare the differences in the physiological effects of villar versus crypt cell contraction. While acute effects of villar cell contractility are largely restricted to non-cell autonomous responses, local effects are much more apparent when contractility is increased in cells of the crypt. The high sensitivity of these cells to intracellular contractility has not been previously demonstrated and suggests that crypt cells are more protected from mechanical forces compared to villar cells. This may be due to the vastly different environments that these cell types are subjected to, despite them being in the same tissue. While villar cells are subject to the open lumen, crypts are enveloped in a mesenchyme and matrix environment that carefully regulates its mechanical and signaling niche. This is exemplified by the fact that intestinal cells cultured under different mechanical conditions can affect their morphogenesis and cell fate [17,40]. Work in the skin epithelium revealed that a decrease in contractility resulted in progenitor dysfunction, including DNA damage, suggesting that progenitor cells may need to finely tune contractility as either increases or decreases can be detrimental to their proliferation [41]. Further work is needed to better understand the mechanoprotective

nature of the intestinal stem cell niche, as well as the physiological mechanisms that underly the tissue differential response to actomyosin contractility.

## Methods

### Ethics statement

All animal work was approved by Duke University's Institutional Animal Care and Use Committee.

### Mice

All mice were maintained in a barrier facility with 12-hour light/dark cycles. Mice were genotyped by PCR and both males and females were analyzed. TRE-Arhgef11$^{CA}$ were generated in the lab as previously described [24]. Other animal strains used in this study were Villin-rtTA (Jackson Laboratories, [42]), VillinCre (Jackson Laboratories [43]), and Rosa-rtTA3 (Jackson Laboratories). For drug rescue experiments, mice were injected with Y-27632 dihydrochloride (ApexBio A3008-50) (10 mg/kg), at 8 hours after doxycycline chow.

### Antibodies

The following antibodies and stains were used: Alexa Fluor Rhodamine Phalloidin (Invitrogen Cat.#: R415; 1:50), Hoechst (Invitrogen Cat.#: H3570; 1:10,000), Rat anti-HA (Roche Cat.#: 1186742300; 1:200–1:100), rabbit anti-mouse caspase-3 (R+D systems Cat.#: AF835; 1:100), rat anti-mouse Cd104 (β4-integrin) (BD Biosciences Cat.#: 553745; 1:100), Rat anti-mouse α-18 (gift from Akira Nagafuchi, [25]; 1:100), Rabbit anti- mouse Olfm4 (Cell Signaling Technology, Cat.#:39141T; 1:400), Rabbit anti- mouse SOX9 (Millipore, Cat.#: AB5535; 1:250), mouse anti- Phospho-Histone H2A.X (Invitrogen, Cat.#: MA1-2022; 1:100), Rabbit anti-mouse non muscle myosin heavy chain IIA (Biolegend, Cat.#: 909001; 1:100), Rabbit anti-mouse non muscle myosin heavy chain IIC (Biolegend, Cat.#: 919201; 1:100), Rat anti-mouse Ki67 (Invitrogen, Cat.# 14-5698-80; 1:100), Rabbit anti-mouse GRASP65 (Abcam, Cat.# ab30315; 1:200), Rat anti-mouse E-cadherin (Invitrogen, Cat.# 13–1900; 1:500), Rabbit anti-Pericentrin (Abcam, Cat.# ab4448; 1:100), Rabbit anti-human ZO-1 (Invitrogen, Cat.# 61–7300; 1:100).

### EdU pulse-chase and labeling

EdU was intraperitoneally injected into adult mice at the dose of 10 mg/kg, after 1 day or 2 days, mice were sacrificed for tissue dissection, or the SI shake off experiments. EdU+ cells were detected following protocols of Click-iT EdU Imaging kit (Thermo Fisher C10337).

### Tissue preparation

The jejunal section of small intestine or the proximal half of the colon were harvested and embedded in optimal cutting temperature compound (O.C.T-Sakura 4583), then frozen. All samples were sectioned at 8 or 10μm thickness and stored at -80˚C prior to staining.

### Whole mount tissue preparation

The small intestine epithelium was prepared following previous procedures [18]. Briefly, the jejunal region of the small intestine was isolated and cut into around 2 cm segments, then the lumen was flushed with cold HBSS containing Ca$^{2+}$ and Mg$^{2+}$ and cut longitudinally, then put in HBSS containing 30 mM EDTA. The tissues were incubated and rotated at 37˚C for 20

minutes or 4˚C for 1–2 hours, then were shaken vigorously. The epithelial sheets were collected into a 15 ml conical tube and fixed in 4% PFA overnight at 4˚C.

## Immunofluorescence staining

Sections were fixed with 4% PFA at room temperature for 10 minutes, washed 3 times with PBST (PBS containing 0.2% Triton X-100), then incubated with blocking buffer (3% BSA with 5% NGS and 5% NDS) for 30 minutes. Sections were then incubated in primary antibodies diluted in blocking buffer for 15 minutes at room temperature or overnight at 4˚C, washed 3 times in PBST, and incubated in secondary antibodies and Hoechst for 30 minutes, washed 3 times in PBST, and mounted in the anti-fade buffer (90% glycerol in PBS plus 2.5 mg/ml p-Phenylenediamine).

For whole mount immunofluorescence staining, after shaking off the SI epithelium sheet, tissue were collected into a 15 ml conical tube, washed using PBS twice, then fixed using 4% PFA overnight at 4˚C or room temperature for 1–2 hours, then washed three times in PBST, blocked for 30 minutes at room temperature and then incubated in the primary antibodies for 1 hour, washed 3 times in PBST, incubated the secondary antibodies for 30 minutes, washed again 3 times in PBST. Finally the tissue were mounted on slides in a region circled by VALAP using anti-fade buffer.

## Imaging, Quantification and Statistics

Section or whole mount staining slides were imaged on a Zeiss AxioImager Z1 microscope with Apotome.2 attachment, PlanAPOCHROMAT 20X/0.8 objective or a Zeiss 780 upright confocal with a 20X/0.8 Plan-Apochromat objective and acquired using Zen software (Zeiss). Images were analyzed using FIJI software. Quantifications of fluorescence intensity of cortical F-actin, Myosin IIC and α-18 were measured for their plot profiles by drawing lines across the junction, then aligned the maxima and trimmed the ends to yield the final plot. Measurements were taken from cells located in the lower third region of each villus, or in the mid-crypt region depending on the analysis. Quantification of Ki67+ cells in Control, Arhgef11$^{Villi}$ and Arhgef11$^{Crypt}$ mice were performed by measuring numbers of Ki67+ cells and total cells per crypt in cross section. Quantification of Olfm4 or Sox9 expression domains were measured by starting at the crypt base and measuring to the top of the cell where antibody was last detected. Identification of whether cells were located within the villar or crypt compartment was identified by tissue morphology. All statistical analysis was performed using GraphPad Prism 5 software and Microsoft Excel. Data were judged to be statistically significant when p value < 0.05 by two-tailed paired or unpaired Student's t test, asterisks denote statistical significance (ns = not significant, *, p < 0.05; **, p < 0.01, ***, p < 0.001, ****, p < 0.0001), as described in individual figure legend. Where no significance is indicated, p values were > 0.05.

## Supporting information

**S1 Fig. Arhgef11Villi expression results in a local increase in adhesions junction tension.** (A and B) Immunofluorescence staining on small intestine sections of α-catenin α18 (red) and HA (green). Note that α18 staining is highest in HA+ cell in B'. Scale bar 50μm. (C and D) Quantification of lateral junction α18 intensity in (C) villar and (D) crypt cells. Data are mean ±SEM. (E and F) Average ratio of the junctional and cytoplasmic fluorescence values in (E) villi, p = 0.0254, ordinary one-way ANOVA, n = 24 cells for control and n = 20 for HA- cells and n = 18 HA+ cells for Arhgef11Villi and (F) crypts p = 0.809, unpaired t-test, n = 48 cells for control and n = 47 for Arhgef11Villi from 3 mice per genotype. (G) Immunofluorescence staining of MyoIIA (green) and HA (red) in Arhgef11Villi villar intestinal epithelia. Scale bar

15μm. (H) Quantification of lateral junction MyoIIA intensity. Data are mean ±SEM.
(TIF)

**S2 Fig. Increasing junctional actin contractility deforms intestinal cell shape as well as the apical surface without affecting cell polarity.** (A and C) Immunofluorescence staining of MyoIIC (green) and HA (red) in control (A) and Arhgef11Villi (C) villar intestinal epithelia. Scale bars for A and C, 20 μm. (B and D) Traces of representative villar epithelial cell shapes in control (B) and Arhgef11Villi (D) HA-(green) and HA+(red) sections. (E and F) Immunofluorescence staining of GRASP65 (green) and HA (red) in control (E) and Arhgef11Vill (F) epithelial sections. Dotted lines mark the basement membrane. Scale bar for E and F, 40 μm. (G) Immunofluorescence staining of ZO1 (green) and HA (red) in Arhgef11Villi villar intestinal epithelia. (H) Staining of HA (green) and E-cadherin (red) in Arhgef11Villi villar intestinal epithelia. (I) Staining of HA (green) and pericentrin (red) in Arhgef11Villi villar intestinal epithelia. Scale bar for G-I 15μm. Dotted lines mark the basement membrane.
(TIF)

**S3 Fig. Short term increased contractility in villar cells does not affect organismal health.**
(A) Ratio of weight change after one and two days of doxycycline induction for control and Arhgef11Villi, n = 3 animals for each genotype. (B) Quantification of small intestine length of control and Arhgef11Villi animals in cm.
(TIF)

**S4 Fig. Effects of increased epithelial cell contractility on the large intestine.** (A-B) Immunofluorescence images of (A) control and (B) ArhgefVilli large intestine sections stained with F-actin (green), β4-integrin (red), and HA (white). Dashed lines denote individual crypts. Scale bar 50μm. (C) Quantification of colon crypt length in microns. For control n = 62 crypts from 3 animals. For ArhgefVilli, n = 52 crypts from 3 animals. p = 0.1166, paired t-test. (D-E) Immunofluorescence images of (D) control and (E) ArhgefCrypt large intestine sections stained with γ-H2aX (green) and HA (red). Dashed lines denote individual crypts. Scale bar 50μm. (F) Quantification of percentages of γ-H2aX positive crypt cells after 12 hours of dox induction n = 4 animals (44 crypts) for control and 4 animals (68 crypts) ArhgefCrypt. p = 0.0961, paired t-test.
(TIF)

**S5 Fig. The non-cell autonomous effects of villar contractility are reversible.** (A)Timeline of experiment. (B) Whole mount immunofluorescent images of control and ArhgefVilli intestinal epithelia stained with HA (green) and phalloidin (red). Scale bar 50μm. (C) Quantification of villi length in microns. For control n = 3 animals (33 villi). For ArhgefVilli, n = 3 animals (43 villi). p = 0.5715, paired t-test. (D) Quantification of crypt length in microns. For control n = 3 animals (109 crypts). For ArhgefVilli, n = 3 animals (134 crypts). p = 0.5710, paired t-test.
(TIF)

**S6 Fig. Stem cell number remains unchanged upon increased villar cell contractility.** (A-B) Immunofluorescence images of (A) control and (B) ArhgefVilli intestine sections stained with HA (red) and Olfm4 (green) Scale bar 50μm. (C) Number of Olfm4-positive cells per crypt. For control n = 63 crypts from 3 animals. For ArhgefVilli n = 55 crypts from 3 animals. p = 0.2617, paired t-test. (D) Average domain of the expression of Olfm4 in crypt base stem cells. For control n = 78 crypts from 3 animals. For ArhgefVilli n = 63 crypts from 3 animals. p = 0.9190 paired t-test (E) Average domain of the expression of Sox9 in crypt base stem cells. For control n = 62 crypts from 3 animals. For ArhgefVilli, n = 52 crypts from 3 animals.

p = 0.8316, unpaired t-test.
(TIF)

**S7 Fig. There is no change in the proportion of goblet cells in Arhgef11Villi mice.** s(A-B) Immunofluorescence staining of Mucin2 (green) and HA (red) in control (A) and Arhgef11-Villi (B) villar intestinal epithelia. Scale bar 50 μm. (C) Quantification of the percentage of Mucin2 positive cells out of total villar epithelial cells counted. For control and Arhgef11Villi n = 3 animals (900 cells per animal). p = 0.3624, paired t-test.
(TIF)

**S8 Fig. Increased crypt cell contractility effects nuclear morphology.** (A-D) Immunofluorescent images of (A,C)control and (B,D)ArhgefCrypt crypt sections stained with (A-B) F-actin (red) or (C-D)Hoechst only (white). Dashed lines denote individual crypts. Scale bar 50μm.
(TIF)

**S1 File. Raw values for main figure graphs.**
(XLSX)

## Acknowledgments

We thank members of the Lechler Lab for comments on the manuscript, and Julie Underwood for expert care of the mice. We thank Deborah Gumucio, University of Michigan for Villin-Cre, Ben Stanger, University of Pennsylvania for Villin-rtTA, Scott Lowe, Sloan Kettering Cancer Center and Lukas Dow, Weill Cornell Medicine for Rosa-rtTA mice.

## Author Contributions

**Conceptualization:** Wenxiu Ning, Terry Lechler.

**Data curation:** Taylor Hinnant, Wenxiu Ning.

**Formal analysis:** Taylor Hinnant, Wenxiu Ning.

**Funding acquisition:** Wenxiu Ning, Terry Lechler.

**Investigation:** Taylor Hinnant, Wenxiu Ning.

**Methodology:** Taylor Hinnant, Wenxiu Ning.

**Project administration:** Terry Lechler.

**Resources:** Terry Lechler.

**Supervision:** Terry Lechler.

**Validation:** Wenxiu Ning.

**Visualization:** Taylor Hinnant.

**Writing – original draft:** Taylor Hinnant, Wenxiu Ning, Terry Lechler.

**Writing – review & editing:** Taylor Hinnant, Wenxiu Ning, Terry Lechler.

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
