## [Decision Letter · Decision Letter 0]

28 Sep 2023

Dear Dr Lechler,

Thank you very much for submitting your Research Article entitled 'Compartment specific responses to contractility in the small intestinal epithelium' to PLOS Genetics.

The manuscript was fully evaluated at the editorial level and by independent peer reviewers. The reviewers appreciated the attention to an important problem, but raised some substantial concerns about the current manuscript. Based on the reviews, we will not be able to accept this version of the manuscript, but we would be willing to review a much-revised version. We cannot, of course, promise publication at that time.

If you decide to revise the manuscript for further consideration at PLOS Genetics, please aim to resubmit within the next 60 days, unless it will take extra time to address the concerns of the reviewers, in which case we would appreciate an expected resubmission date by email to plosgenetics@plos.org.

We are sorry that we cannot be more positive about your manuscript at this stage. Please do not hesitate to contact us if you have any concerns or questions.

Yours sincerely,

Ophir Klein

Guest Editor

PLOS Genetics

Gregory Barsh

Editor-in-Chief

PLOS Genetics

Reviewer's Responses to Questions

**Comments to the Authors:**

Reviewer #1: uploaded as attachment

Reviewer #2: Review is uploaded as an attachment

**Have all data underlying the figures and results presented in the manuscript been provided?**

Reviewer #1: Yes

Reviewer #2: Yes

PLOS authors have the option to publish the peer review history of their article (what does this mean?). If published, this will include your full peer review and any attached files.

Reviewer #1: No

Reviewer #2: No

Reviewer 1 comments:

This manuscript addresses the interesting question of how epithelial cell mechanics affect tissue function. In particular, the authors focus on how epithelial cell contractility impacts the morphology and homeostasis of the mouse small intestine. While many studies have examined similar questions in vitro, this study uses a new mouse strain, recently developed by the same group, to induce epithelial contractility and study its effects in vivo. The ability to begin addressing these questions in vivo is a major advancement over previous works, as it enables the study of these cellular behaviors in their natural and complex microenvironment. However, prior to publication in PLOS Genetics, this study would require and greatly benefit from several validation experiments to fully support its conclusions and increase its appeal beyond an intestine-specific audience.

**Major comments:**

The data presented herein convincingly show that the overexpression of the Arhgef11 results in morphological and functional differences in the intestine. However, the study lacks appropriate evidence to connect these phenotypes directly to changes in contractility or increased tension as claimed, as opposed to the result of other effects independent of these factors.

1.     To convincingly show that Arhgef11-CA expression induces contractility in intestinal epithelial cells, the authors should use functional cell, organoid, or explant-based assays (e.g.’s gel contraction/tissue compaction, laser ablation of junctions, genetically encoded tension sensors) as opposed to relying solely on the expression of a few IF markers.

2.     To convincingly demonstrate that the phenotypes observed are due to cell contractility or tension heterogeneity within the epithelium, the authors should 1) test whether these phenotypes (in both crypt and villus domains) are recapitulated in 2D or 3D intestinal organoids isolated from these mice, and if so whether they can be rescued by ROCK inhibition (e.g. Y27632 or Thiazovivin). Additionally, they should test whether activation of contractility in vitro through Calyculin A treatment or other means (e.g. lentiviral transduction of other contractility inducers) recapitulates the observed phenotypes.

3.     “The changes in contractility appear to be quite local and we have no evidence that they are transmitted through multiple cells.” This claim should be investigated in more detail by quantifying the junctional MyoII C etc. as well as phosphorylated myosin light chain in cells adjacent and within the vicinity of the ArhgefCA-expressing villus cells. From some of the images (e.g. 1I and S2C), it appears that there might be subtle differences that could be more obvious when quantified and would help explain the non-cell-autonomous effects – perhaps one of the most exciting findings of this manuscript.

**Minor comments:**

1.     The timing of dox induction and tissue collection is unclear for some of the experiments. The manuscript would benefit from adding the experimental timelines to all figures.

2.     Where along the crypt-villus axis were the line profiles for quantification drawn? Was this specifically for cells along the villus sides or at the tips? The locations could themselves have heterogeneity between them in control samples, so an important control would be to measure the line profiles of Actin/MyoIIC at different positions along the villus or at least clearly state where the measurements are made.

3.     Are there changes to villus morphology in the Arhgef11-Crypt mice? It would be assumed that the villi would undergo atrophy but this should be more clearly shown.

4.     Are the Arhgef11-crypt phenotypes specific to the small intestine, or is the expression/phenotype also observed in the colon?

5.     Multiple misspellings of “Arhgef”

6.     Several of the conclusions are overstated and should be adjusted accordingly or validated by experiments.

·       “for the first time, reveals the mechanical sensitivity of small intestinal epithelial stem cells”

Multiple studies, included some cited here, have shown this (e.g. PMIDs: 31019299, 31019299, 31019299, 31019299)

·       “Regardless, these data indicate that the crypt hyperplasia, in response to increased actomyosin contractility in villar cells, is driven by transit amplifying cell hyperproliferation and not an expansion of stem cells.”

o   This is not convincing from the graph of Sox9 expression alone. To make this claim the authors should look at the proportion of more bona fide ISC markers, such as Lgr5 and Olfm4.

“Together these data demonstrate that mechanical deformation induces DNA damage leading to apoptosis of these cells”Not directly tested that this is causal

Reviewer 2 comments:

This manuscript by Hinnat *et al. *reports on an interesting topic related to the crosstalk between the differentiated and proliferative compartments in the intestine from a contractility point of view. The authors genetically induced a constitutively active HA-tagged fragment of Arhgef11 to promote contractility in cells where recombination takes place. They found that induction in villus cells, leads to morphological changes cell autonomously, while the crypt compartment undergoes a robust hyperproliferative phenotype. Interestingly, when Arhgef11 gain-of-function is restricted to the crypts, cells undergo cell death, leading a decline in body weight and intestinal length by 24hr after Dox induction. The manuscript is interesting. However, several experiments need to be included to support the strong conclusions that the authors draw from their current data.

Major comments:

The authors must elaborate on the reasons for actomyosin gain-of-function experiments in the introduction. It needs to be clarified what was the hypothesis that the authors were trying to test in this study. For example, the following sentence is vague and needs clarification on what function remains unclear: “Despite the importance of actomyosin-mediated contractility in intestinal development and homeostasis, the effects that contractility has on cell and tissue function remain unclear.”

The authors should refrain from statements that are not accurate, such as: “This study highlights distinct, compartment specific responses to cell contractility and, **for the first time**, reveals the mechanical sensitivity of small intestinal epithelial stem cells.” There are published studies testing “mechanical sensitivity” in the crypt compartment, including stem cells. To name a few: Pentinmikko et al. 2022; Pérez-González et al. 2021; Yang et al. 2021; Sumigray et al. 2018).

The authors should explore the expression of NMIIA, which is also expressed in the intestinal epithelium, and they are strongly encouraged to stain for pMCL, which indicates activated myosin and contracting stress fibers.

A more robust panel of cell polarity markers needs to be assessed to determine changes in cell polarity.

An analysis of cell death in villi is needed. Even though the authors show that villus length is maintained in mutant mice, the increase in crypt proliferation might result in proper villus length by replacing lost villus cells. Their conclusion that the only observed phenotype in villus cells is the change in morphology will be further supported if there’s no difference in cell death in villus cells.

The authors cannot conclude that Arhgef gain-of-function is restricted uniquely to villus cells since they show the presence of HA+ cells high up in the crypt compartment. Additionally, the authors’ conclusion that there is a non-cell autonomous effect from villus cell contraction cascading to the proliferative compartment is intriguing and relevant. Still, it is not supported by their current data. Their manuscript and conclusion will require a villus-specific driver, for example, K20CreER, to drive the expression of their allele specifically and uniquely in differentiated villus cells.

Like point #4, the authors need to use a more robust list of markers to support their conclusion that the TA progenitors, not the stem cells, are the cell population driving the hyperproliferative phenotype. Lgr5, Olfm4, Math1, Notch1, etc., markers should be included.

Are there any defects in cell differentiation after Dox induction?

The authors claim that recombination in the Arhgef11^Villi^ mouse model occurs only in villus cells and cells at the villus-crypt junction, which are differentiated cells lost within 3-5 days in the small intestine. Thus, is the phenotype reversible? After the mosaic recombined differentiated cells are replenished by new non-recombined cells, do the villus and crypt phenotypes back to homeostasis?

Have the authors tested increasing the Dox induction paradigm of the Arhgef11^Villi^ mouse model for longer than 3 days? This is an interesting experiment to carry out to support their conclusion that the crypt phenotype is driven by TA cells. As TAs don’t self-renew, a likely outcome would be that these cells exhaust themselves, and the crypt phenotype should resolve back to homeostasis.

As in point #3, the authors in Fig. 4 need to stain for pMCL to show increased contractility.

The authors should adjust their conclusions to reflect their data. For the authors to conclude that “crypt progenitor cells are more sensitive to increased actomyosin contractility” they need to induce actomyosin contractility in the progenitor cells. Their current Arhgef11^crypt^ genetic tool is active throughout the crypt. Thus, the role of the distinct crypt cell populations (stem cells, Paneth cells, secretory progenitors, and absorptive progenitors) cannot be disentangled.

Is the basement membrane of the injured crypts impaired?

The enthusiasm for the manuscript was significantly lowered due to the absence of a physiologically relevant model that would mimic the conditions that the authors are inducing genetically– i.e., increased actomyosin-induced contractility.

Other comments:

This reviewer wants to commend the authors for the whole mount analysis and the beautiful, high-quality images.

Statistics should be done on the average “measured unit” per mouse and not using each data point. For example, the statistical analysis of Fig. 2C-F needs to be reanalyzed using the average villi length –for C– or the average crypt length –for F– per mouse instead of doing the statistics using each villus (n=20-40) or each crypt (n=51-64) data point. The correct statistical analysis that should be applied across the manuscript is illustrated in Fig. 2H.

Related to point #9. The authors are encouraged to show the distribution of the average “measured unit” per mouse as individual values using graphs such as Fig. 2H and avoid using bar graphs that do not show the individual values, such as Fig. 2I.

Can the authors choose two representative images in Fig. 4I instead of the current low-magnification image, which does not get their point across?

In Fig. 4D-E, the authors should add a reference point on where these images are taken along the crypt axis, i.e., are the cross sections at the +4 position, mid-crypt, or below the crypt-villus hinge? This information would be helpful since the lumen of the crypt changes in size as you move up the crypt axis. Further, these panels will significantly improve if the authors use their whole mount tissue for 3D reconstruction of the control and mutant crypts.

It needs to be clarified what section of the small intestine was used for their analysis. In the methods section, they mentioned the “medial” section of the small intestine. Does this mean Jejunum? Either use the duodenum, jejunum, or Ileum terminology or clarify how far from the pylorus or pyloric sphincter the analysis was made. This is relevant because of the transcriptional, cellular, and morphological differences along the proximal-distal small intestine.

---

## [Decision Letter · Decision Letter 1]

22 Feb 2024

Dear Dr Lechler,

Thank you very much for submitting your Research Article entitled 'Compartment specific responses to contractility in the small intestinal epithelium' to PLOS Genetics.

The manuscript was fully evaluated at the editorial level and by independent peer reviewers. The reviewers appreciated the attention to an important topic but identified some concerns that we ask you address in a revised manuscript.

We therefore ask you to modify the manuscript according to the review recommendations. Your revisions should address the specific points made by each reviewer.

Yours sincerely,

Ophir Klein

Guest Editor

PLOS Genetics

Gregory Barsh

Editor-in-Chief

PLOS Genetics

Reviewer's Responses to Questions

**Comments to the Authors:**

Reviewer #1: uploaded as attachment

Reviewer #2: Review is uploaded as an attachment

**Have all data underlying the figures and results presented in the manuscript been provided?**

Reviewer #1: Yes

Reviewer #2: None

PLOS authors have the option to publish the peer review history of their article (what does this mean?). If published, this will include your full peer review and any attached files.

Reviewer #1: No

Reviewer #2: No

---

## [Editor Report · Decision Letter 2]

7 Mar 2024

Dear Dr Lechler,

We are pleased to inform you that your manuscript entitled "Compartment specific responses to contractility in the small intestinal epithelium" has been editorially accepted for publication in PLOS Genetics. Congratulations!

Yours sincerely,

Ophir Klein

Guest Editor

PLOS Genetics

Gregory Barsh

Editor-in-Chief

PLOS Genetics

Comments from the reviewers (if applicable):

**Data Deposition**

http://datadryad.org/submit?journalID=pgenetics&manu=PGENETICS-D-23-00884R2

**Press Queries**

---

## [Editor Report · Acceptance letter]

19 Mar 2024

PGENETICS-D-23-00884R2 

Compartment specific responses to contractility in the small intestinal epithelium 

Dear Dr Lechler, 

We are pleased to inform you that your manuscript entitled "Compartment specific responses to contractility in the small intestinal epithelium" has been formally accepted for publication in PLOS Genetics! Your manuscript is now with our production department and you will be notified of the publication date in due course.

With kind regards,

Anita Estes

PLOS Genetics

On behalf of:
